# The Resource Potential and Zoning Evaluation for Deep Geothermal Resources of the Dongying Formation in Tianjin Binhai New Area

**Jiulong Liu** [1,*]**, Shuangbao Han** [1]**, Hong Xiang** [1,*]**, Dongdong Yue** [1] **and Fengtian Yang** [2]

1   Center for Hydrogeology and Environmental Geology Survey, China Geology Survey,
    Tianjin 300304, China; shuangbaohan@126.com (S.H.); yuedongdong@mail.cgs.gov.cn (D.Y.)
2   Key Laboratory of Groundwater Resources and Environment, Ministry of Education, Jilin University,
    Changchun 130021, China; yangfengtian@jlu.edu.cn
*   Correspondence: julon0920@163.com (J.L.); linxi.880130@163.com (H.X.)

**Abstract:** The Dongying geothermal resources are an important part of the deep geothermal resources in Tianjin Binhai New Area, and these resources will determine whether the demand target can be met. There is an urgent need to find favorable target areas for Dongying geothermal resources and to develop Dongying geothermal resources safely, stably and efficiently. The recoverable potential of the Dongying geothermal reservoir in different positions is calculated by simulating and predicting the water level, which is as an important index of zoning evaluation. By using the analytic hierarchy process (AHP), the comprehensive indexes of geothermal exploration and development regionalization are quantified. A grade-two evaluation system, which considers the development constraints, has been established for the delineation of deep geothermal resources' exploration and development prospect target areas. The zoning evaluation results show that the excellent prospect target area of Dongying geothermal resources, for their exploration and development, is 314.33 km$^2$, the general prospect target area is 745.77 km$^2$, and the bad prospect target area is 879.31 km$^2$. The quantitative zoning evaluation method can provide references for the optimization of the exploration and development target area of deep geothermal resources with low prospecting accuracy in key areas of China.

**Keywords:** zoning evaluation; prospect target area; resource potential; Dongying geothermal resources; Tianjin Binhai New Area





## 1. Introduction

As a form of renewable green energy, geothermal energy is abundant, stable and safe. With the goal proposed of the "double carbon" target, geothermal property has been subject to rapid development, and geothermal energy has experienced a significant increase in the efforts of energy structure adjustment. According to incomplete statistics, by the end of 2020, China's geothermal utilization was equivalent to about 40 million tons of standard coal (only 4.5% of non-fossil energy resources) and the deep geothermal heating area is up to 580 million square meters [1]. The geothermal resources in the Bohai Bay basin are widely distributed [2]. With the rapid development of Tianjin Binhai New Area, the demand for geothermal energy is increasing. As a result, this area has become one of the most important mid–deep hydrothermal geothermal development zones. The Dongying geothermal resource is an important part of the mid–deep hydrothermal geothermal resources in Tianjin Binhai New Area, and both the amount of geothermal fluid and the supply relationship have become important; together, these will determine whether the demand target can be met [3]. In order to develop deep geothermal resources safely, stably and efficiently, it is necessary to carry out zoning of deep geothermal resources.

With the increase in exploration and development of deep geothermal resources in recent years, early studies have provided significant knowledge about the zoning evaluation of medium–deep geothermal resources. In 1995, Dunshi Yan, Yingtai Yu and others carried out significant work on the geothermal distribution, type division, resource quantity calculation, development and utilization evaluation in the oil and gas region of Beijing–Tianjin–Hebei [4]. Wenjing Lin used different evaluation methods to evaluate the potential for the different types of shallow geothermal energy, hydrothermal geothermal resources and dry hot rock resources in China [5]. Guiling Wang analyzed the current situation, along with the economic and environmental benefits of the development and utilization of geothermal resources [6]. Zhonghe Pang proposed an index system for evaluating the mining conditions of deep geothermal energy resources, according to which they assigned values to each index via expert scoring, and then quantitatively calculated and evaluated the development difficulty of deep geothermal energy resources using fuzzy mathematics [7]. Zongming Liu constructed an evaluation system of 61 geological condition indicators, including basic geology conditions, geological environment elements and geological resource elements, in order to evaluate the urban geology conditions of Beijing [8]. Based on the development characteristics and utilization direction of karst fissure geothermal resources in Shandong Province, Zhongxian Gao established a selection evaluation method using resource and market conditions [9]. Shengtao Li established a site-selection evaluation index system for dry hot rocks' exploration, which includes four aspects: resources, technology, safety and economy [10]. According to the project selection, project establishment, construction and operation stages, Guoyong Liu proposed a set of evaluation systems for hydrothermal geothermal resources in the middle and deep layers of the sedimentary basin [11].

The evaluation methods in these studies include resource volume calculation, numerical simulation and comprehensive evaluation according to physical indicators. These methods lack the quantitative evaluation of the comprehensive index and do not consider either the recoverable potential or the development constraints. Based on that previous research, this paper introduces a comprehensive evaluation method of geothermal geology multi-source information data fusion using GIS. In order to effectively reduce the risk of geothermal resource development and to provide a reliable basis for the planning and management of geothermal resource development and utilization, it is necessary to carry out methods selection and quantitative evaluation of deep geothermal resource zoning. This paper takes the Dongying geothermal resource in Tianjin Binhai New Area as the research object. By using the analytic hierarchy process (AHP), the index of deep geothermal resource exploration and development zoning is quantified, and the prospect target areas of deep geothermal resource exploration and development are delineated on a regional scale using the quantitative zoning evaluation method. The purpose of this exercise is to construct the zoning evaluation method for deep geothermal resources and to provide references for the exploration and development of middle–deep geothermal resources with low prospecting accuracy in key regions of China.

## 2. Geological Setting

### (1) Geological tectonic characteristics

The Bohai Bay basin is one of the important medium–deep hydrothermal–geothermal development areas in China [12] and is rich in geothermal resources. This area largely consists of the huge Meso-Cenozoic depression and is mostly low-lying land and lacustrine marsh, with a sedimentary thickness of 1500~5000 m [13]. This study focuses on the Dongying formation geothermal reservoir in Tianjin Binhai New Area. The study area is located in the north of the Huanghua depression, which is a grade III structural unit of the Bohai Bay basin. The Huanghua depression is bounded by the Cangxian uplift to the west and the Bohai Sea to the east, the Chengning uplift to the southeast and the Yanshan Mountains fold belt to the north [14]. The major faults developed in the region include the Cangdong fault, the Hangu fault, the Haihe fault, the Zengfutai fault and the Beidagang

fault. These NE-trending faults cut off from each other in the NW and NW directions, and they form some secondary structural units: the Ninghe swell, Beitang sag, Banqiao sag and Qikou sag [15] (Figure 1).

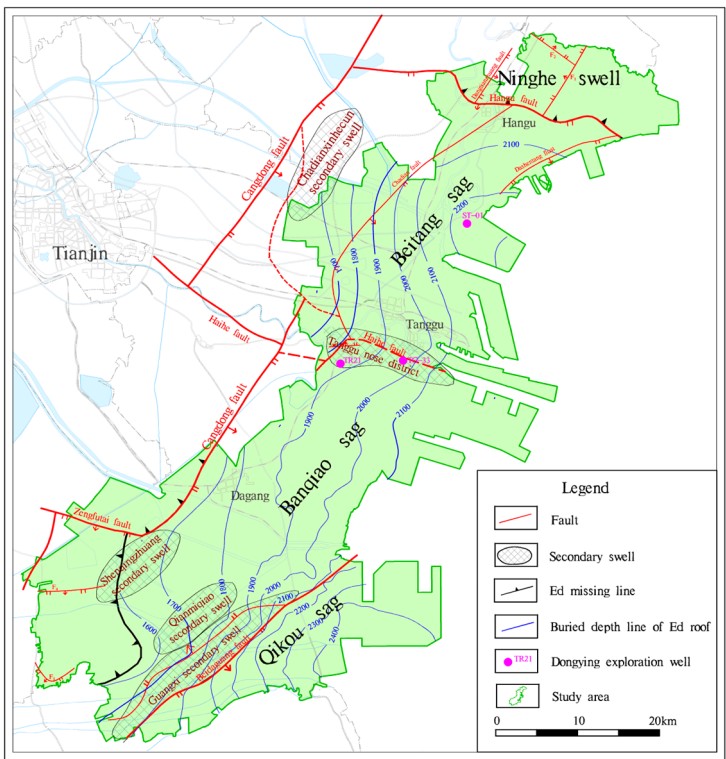

**Figure 1.** Tectonic map of the study area.

The Paleogene Dongying formation in the Tianjin coastal regions is widely distributed to the east of the Cangdong fault but it is absent to the west of the Cangdong fault [16]. The Neogene and Quaternary deposits are overlying the Paleogene Dongying formation in Tianjin Binhai New Area. The characteristics are a loose structure, small density, large thickness and poor thermal conductivity [17], which make the heat conduction in the deep crust long-lasting. It is forming a good thermal reservoir cap rock.

**(2) Geothermal reservoir characteristics**

According to the drilling data [18,19], the roof buried depth of the Dongying geothermal reservoir is 1500~2300 m (Figure 1), the thickness is 100~660 m, the sand–mud ratio is 35~38%, the porosity is 20~35%, and the permeability is 200~1000 mD. The roof buried depth of the Paleogene Dongying geothermal reservoir is 1500~2200 m, and the permeability is 200~900 mD to the south of the Haihe fault. The roof buried depth of the Paleogene Dongying geothermal reservoir is 1500~2200 m, and the permeability is 1600~2300 mD to the north of the Haihe fault. In the lower part of the Dongying formation, greyish-green mudstone is interbedded with sandstone and glutenite, with a thickness of about 300 m and a porosity of 20~28%. The upper part of the Dongying formation is composed of mainly sandstones, ranging from gray-green to mottled purplish-red mudstones, and greyish-white mixed sandstones, lithic arkose and glutenite; its thickness is about 200 m and its porosity is 22~35%. The middle part is dominated by mud shale with a poor water-bearing condition [20].

Based on the lithology combination rule of Dongying exploration wells, combined with the previous research results [21,22], the Dongying deposition in the study area is mainly a set of terrigenous clastic deposition with mudstone and siltstone [23], including feldspar and quartz, which are angular or sub-angular and have a poor degree of rounding. From the geophysical profile curve (Figure 2), which can reflect sedimentary sequence changes [18], combined with the depositional characteristic of the low sand–mud ratio and

the more indicative marine fossils, it can be seen that the terrigenous clastic of the Paleogene Dongying formation in the study area is delta deposition controlled by rivers [24,25]. Delta siltstones provide favorable hydrogeology conditions for groundwater enrichment and runoff, especially the estuarine sand bars and front sheet sand bodies of river-controlled deltas, which have good water-storage properties [26].

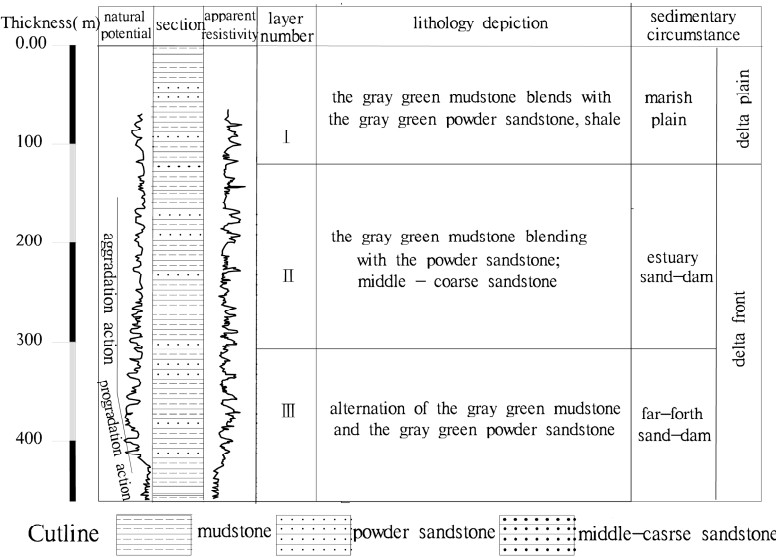

**Figure 2.** Sedimentary faces analysis of Paleogene Dongying formation.

Taking Dongying exploration well TR21 as an example [18], the water output of the single well is about 40 m³/h, the water temperature is 73 °C, the TDS of geothermal fluid is 2993.2 mg/L, and the hydrochemical type is Cl·HCO$_3$-Na. The core analysis results of a 2225~2226.8 m coring section show that the sandstone is gray-green siltstone with a fine grain structure and good diagenesis. Meanwhile, the pore diameter is 0.05~0.1 mm, the observed porosity is 30~35%, and the impurity content is less than 15%. The porosity of the geothermal reservoir is large, but its shale content is high (Table 1).

**Table 1.** The porosity and permeability statistical table of Dongying well TR21.

| Section | Buried Depth (m) | Thickness (m) | Porosity (%) | Permeability ($\times 10^{-3}$ μm²) | Clay Content (%) |
|---|---|---|---|---|---|
| Ed I | 1950.1~1963.1 | 13.0 | 27.51 | 625.25 | 23.27 |
| | 1978~1984 | 6 | 26.64 | 480.03 | 21.21 |
| | 1989.4~1996.7 | 7.3 | 23.35 | 218.93 | 21.86 |
| Ed II | 2073.3~2090.1 | 16.8 | 31.41 | 908.16 | 14.54 |
| | 2202.2~2207.6 | 5.4 | 31.02 | 812.55 | 12.54 |
| Ed III | 2213.0~2220.0 | 7 | 29.98 | 725.69 | 13.22 |
| | 2249.7~2256.1 | 6.4 | 25.94 | 389.88 | 16.15 |
| | 2272.3~2278.8 | 6.5 | 26.04 | 406.29 | 13.19 |
| | 2284.3~2312.6 | 28.3 | 23.41 | 275.69 | 19.08 |
| | 2320.2~2324.9 | 4.7 | 26.41 | 421.98 | 12.89 |

**(3) Resource potential conditions**

In this calculation, the mapping virtual well method is used to generalize the water-resisting boundary into an infinite boundary, and any other lateral boundary is treated as an infinite boundary [27]. Here, the mapping method is used to deal with the problem of the water-resisting boundary, which makes the boundary extend infinitely. The characteristics of the reservoir media are generalized as homogeneous isotropy, and the migration law of

geothermal fluid is basically in accordance with Darcy's law, along with meeting the Theis model of two-dimensional groundwater seeping.

According to GB11615-2010 [28], the available-reserves calculating period of the medium- and low-temperature geothermal fields is 100a. Taking into account the head pressure of the Dongying geothermal reservoir and the actual working capacity of the mining equipment at this stage, the maximum static water level depth should not exceed 175 m. Based on three existing geothermal production wells (TG-33, TR21 and ST-01), and combined with stratigraphic conditions, structural characteristics and regional municipal planning requirements in the blank area of geothermal development, the virtual wells are designed in the area. Different mining capacities are input into the model, and the corresponding buried depth of the water level at each control point is calculated. The mining well with the largest buried depth of water level is found, and the calculated buried depth of water level is compared with the maximum allowable buried depth of water level, to find the most suitable water level buried depth. At this time, the amount of exploitation is the corresponding amount of fluid recoverable.

Taking into account the actual production situation and the recharge ability test of the existing recharge wells, the exploitation potential of the geothermal fluid is calculated according to the two options of the single-well production model and the double-well production and irrigation model of the recharge rate of 30%.

Under the single-well production model, a total of 29 simulated wells (virtual wells) were designed. The Cangdong fault, the Hangu fault and the missing line of the Dongying formation in the study area were generalized to the water-resisting boundary [29]. According to the principle of virtual well mapping, the mapping wells were included in the model calculation. The calculated results show that the recoverable capacity of the Dongying geothermal fluid in the study area is $233.6 \times 10^4$ m$^3$/a under the single-well production model (Table 2).

**Table 2.** The recoverable capacity of geothermal fluid under the single-well model.

| Reservoir | Production Capacity of Single Well/m$^3$/d | Production Volume of the Whole Area /10$^4$ m$^3$/d | Maximum Water Level Depth/m | Minimum Water Level Depth/m | Recoverable Capacity/10$^4$ m$^3$/a |
|---|---|---|---|---|---|
| | 180 | 0.576 | 169.0957 | 124.5534 | |
| Ed | 200 | 0.640 | 174.5346 | 128.3976 | 233.6 |
| | 220 | 0.704 | 181.6459 | 134.9912 | |

Under the production and irrigation wells model, a total of 29 simulated production and irrigation wells (virtual wells) were designed. The calculated results show that the recoverable yield of the Dongying geothermal fluid in the study area is $315.36 \times 10^4$ m$^3$/a when the recharge rate of irrigation wells is 30% (Table 3).

**Table 3.** The recoverable capacity of geothermal fluid under the double-well model.

| Reservoir | Production Capacity of Double Well/m$^3$/d | Production Volume of the Whole Area /10$^4$ m$^3$/d | Maximum Water Level Depth/m | Minimum Water Level Depth/m | Recoverable Capacity/10$^4$ m$^3$/a |
|---|---|---|---|---|---|
| | 250 | 0.800 | 167.0524 | 132.8215 | |
| Ed | 270 | 0.864 | 175.1002 | 138.1144 | 315.36 |
| | 290 | 0.928 | 181.4361 | 146.0035 | |

The temperature of the Dongying geothermal reservoir is 60~90 °C, and the average temperature is 75 °C. The Dongying geothermal reservoir in the study area is as a middle–low-temperature hydrothermal reservoir. The available geothermal energy reserves of the Dongying formation can be calculated as 100a. The available thermal power is shown in Table 4, which is less than 50 MW and more than 10 MW, so the reserve level is medium.

**Table 4.** The available heat resource scale of the geothermal fluid.

| Reservoir | Mining Model | Recoverable Capacity /$10^4$ m$^3$/a | Specific Heat Capacity /kJ/m$^3$·°C | Available Amount of Heat Energy/$10^{13}$ kJ | Available Thermal Power/MW |
|---|---|---|---|---|---|
| Ed | single-well model | 233.6 | 4054.6 | 5.825 | 18.47 |
| | double-well model | 315.36 | 4054.6 | 7.864 | 24.93 |

## 3. Evaluation Methods

By selecting scientific evaluation methods, the exploration and development prospect area of geothermal resources is divided into zones, which provides a reliable basis for the planning and management of the middle and deep geothermal resources [30]. Licai Liu and Miaojuan Xu applied the analytic hierarchy process (AHP) to the suitable zoning of shallow geothermal resources for development and utilization [31,32]. The evaluation model of the analytic hierarchy process (AHP) is used to realize the quantization of division index [33]. On the basis of comprehensive analysis of geothermal geology conditions in the study area, this paper draws from the previous technical evaluation experience of suitability zoning of geothermal resources for development and utilization using the analytic hierarchy process (AHP). The analytic hierarchy process (AHP) is a relatively new model developed by professor T.L. Saaty, who was an operational research scientist in the USA in the early 1970s. This method quantifies the decision-maker's experience, and it is convenient to use when the target factors are complex and necessary data are lacking, so it is widely used in practice [34].

(1)    The determination of evaluation indicators

From the perspective of geothermal resource development, geothermal resource target areas should have sufficient geothermal resource quantity, development and utilization potential, and geothermal energy demand [35,36]. The geothermal geology conditions are the key factors of the prospective areas delineating geothermal exploration and development, which are the first-level evaluation index [37]. The development constraint conditions play a direct role in the evaluation, which are the second-level evaluation index. Therefore, the delineation and evaluation system of the Dongying geothermal exploration and development prospect area in the study area is divided into two levels, and the evaluation structural model is shown in Figure 3.

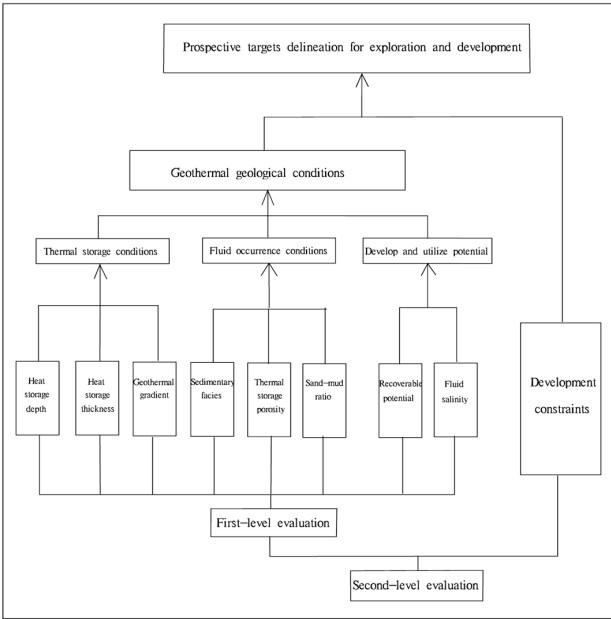

**Figure 3.** Structure diagram of evaluation model.

Since there are few geothermal wells of the Dongying formation in study area, and the data are mainly regional, we chose the following indicators for evaluation.

The heat storage conditions include the buried depth of the geothermal reservoir, the thickness of the geothermal reservoir and the temperature gradient of cap rock. They reflect the heat storage space and heat condition of the geothermal reservoir, and they also affect the economy of well completion.

The fluid occurrence conditions include the sedimentary characteristics of the geothermal reservoir, the porosity development of the geothermal reservoir and the proportion of the sand and mud of the geothermal reservoir, which reflect the water-rich property of the geothermal reservoir and represent its water supply capacity; at the same time, they also affect the reinjection effect of geothermal fluid.

The development and utilization potential include the recoverable potential of geothermal resources' exploitation and the salinity characteristics of geothermal fluid. Based on the hydrogeological parameters of the thermal reservoir and the tectonic boundary, the virtual production wells are arranged, the annual recoverable exploitation capacity of unit drawdown ($m^3$/a·m) of the geothermal wells is calculated by simulating and predicting the water level, and the recoverable potential of the resources is divided. The recoverable potential of geothermal resources' exploitation characterizes the replenishment capacity of geothermal resources. It is related to the sustainable development and utilization of geothermal resources, and it affects whether the demand for geothermal heating can be met.

As for the oil-rich areas of the Dongying formation and natural reserves in the study area, these do not meet the necessary conditions to exploit the geothermal resources. These areas are development constraints in the evaluation.

(2)　The given evaluation score of the interval attribute

Combined with the characteristics of a deep geothermal reservoir, and taking full account of the factors affecting the development and utilization potential of geothermal resources, in order to delineate the favorable target areas of geothermal resources, a comprehensive evaluation index system and evaluation method of geothermal resource target areas in the study area were established [38]. These functioned by quantifying the data used in the evaluation and considering whether they favor the exploration and exploitation of geothermal resources, which was taken as the comparison standard. The importance of evaluation indicators was compared using the 1~9 scale method, and a judgment matrix of importance paired comparison was established. The concrete method was to assign an attribute value to each area in the basic map of each factor index (the more favorable condition for exploration and development of geothermal resources, the higher the score). The evaluation level and interval division were mainly based on empirical values and the division standards in the Geothermal Resource Evaluation Method and Estimation Regulations (DZ/T0331-2020) [39]. The scoring algorithm was derived using linear formulas based on the actual values of the indicators in the study area, as shown in Table 5. Then, the evaluation area was meshed, and the meshing map was overlaid with the indicator image that was assigned a value, and the value of each indicator image was extracted through its spatial analysis function. Thus, the assignment in the graph corresponded to the corresponding grid points.

**Table 5.** Assignment table of the indicators.

| Evaluation Indicators | Grading | Assignment | Evaluation Indicators | Grading | Assignment |
|---|---|---|---|---|---|
| Buried depth of geothermal reservoir (m) | <2000 | 8 | Thickness of geothermal reservoir (m) | <200 | 2 |
| | 2000~2600 | 7 | | 200~400 | 5 |
| | 2600~3200 | 5 | | 400~600 | 7 |
| | >3200 | 3 | | >600 | 8 |

**Table 5.** *Cont.*

| Evaluation Indicators | Grading | Assignment | Evaluation Indicators | Grading | Assignment |
|---|---|---|---|---|---|
| Temperature gradient of cap rock (°C/100 m) | <2.5<br>2.5~3.0<br>3.0~3.5<br>>3.5 | 2<br>5<br>7<br>9 | Sedimentary characteristics | Delta front<br>Delta plain<br>Alluvial fan<br>Shallow lake | 8<br>6<br>5<br>3 |
| Porosity of geothermal reservoir (%) | <15<br>15~20<br>20~25<br>>25 | 3<br>6<br>7<br>9 | Sand–mud ratio (%) | <10<br>10~20<br>20~30<br>>30 | 2<br>4<br>6<br>8 |
| Recoverable potential ($10^4$ m$^3$/a·m) | <4<br>4~6<br>>6 | 3<br>6<br>8 | Salinity of geothermal fluid (mg/L) | <3000<br>3000~10,000<br>>10,000 | 8<br>6<br>4 |

(3)  The quantification of evaluation factors weight

The weight is a quantized value which represents the effect of the lower sub-criteria relative to the upper one [40]. In order to ensure the reliability and credibility of the weight taken by each evaluation index factor, the analytic hierarchy process (AHP) evaluation model was used to calculate the weight of factors [41]. The weight of each evaluation index was evaluated comprehensively by an expert scoring method and the analytic hierarchy process.

The basic principle of the analytic hierarchy process (AHP) model is as follows: for complex social public management problems, the AHP structure model is established and the judgment matrix is constructed. The eigenvalue method can be used to determine the importance ranking weights of various schemes and measures for decision-makers' reference. Using analytic hierarchy process (AHP) modeling follows four steps: (i) establishing the hierarchy model; (ii) constructing judgment matrices in each hierarchy; (iii) a hierarchical single ordering and consistency test; (iv) a hierarchical total ordering and consistency test.

The analytic hierarchy process (AHP) produces a relative value obtained by comparing the advantages of each index, that is, the superiority weight [42,43]. The weight value is determined and tested by the judgment matrix. The consistency ratio that the comparison judgment matrix of the eight evaluation indicators constructed in this evaluation was far less than 0.1, with satisfactory consistency.

According to the requirement of the analytic hierarchy process (AHP), on the basis of the hierarchical subordination of the first-level evaluation system, through statistical and research analysis, using the 1–9 scale method, the comparison matrix is formed by comparing the importance of each factor in the attribute layer and element layer (the more important factor has great influence on the delineation of vision area). Through the calculation, the consistency of the comparison matrix is tested, and the comparison matrix is modified if necessary to achieve acceptable consistency. The importance comparison of the evaluation factors is shown in Table 6.

**Table 6.** The importance comparison table of the evaluation factors.

| 1. Geothermal geology conditions Consistency ratio of judgment matrix: 0.0088 The weight of the overall goal: 1.0000 | | | | |
|---|---|---|---|---|
|  | Heat Storage Conditions | Fluid Occurrence Conditions | Develop and Utilize Potential | $W_i$ |
| Heat storage conditions | 1.0000 | 0.3333 | 0.5000 | 0.1634 |
| Fluid occurrence conditions | 3.0000 | 1.0000 | 2.0000 | 0.5396 |
| Develop and utilize potential | 2.0000 | 0.5000 | 1.0000 | 0.2970 |
| 1.1 Heat storage conditions Consistency ratio of judgment matrix: 0.0176 The weight of the overall goal: 0.1634 | | | | |
|  | Buried depth | Buried thickness | Temperature gradient | $W_i$ |
| Buried depth | 1.0000 | 0.2500 | 0.5000 | 0.1365 |
| Buried thickness | 4.0000 | 1.0000 | 3.0000 | 0.6250 |

**Table 6.** *Cont.*

| | Temperature gradient | 2.0000 | 0.3333 | 1.0000 | 0.2385 |
|---|---|---|---|---|---|

| 1.2 Fluid occurrence conditions Consistency ratio of judgment matrix: 0.0088 The weight of the overall goal: 0.5396 | | | | |
|---|---|---|---|---|
| | Sedimentary characteristics | Porosity | Sand–mud ratio | $W_i$ |
| Sedimentary characteristics | 1.0000 | 2.0000 | 3.0000 | 0.5396 |
| Porosity | 0.5000 | 1.0000 | 2.0000 | 0.2970 |
| Sand–mud ratio | 0.3333 | 0.5000 | 1.0000 | 0.1634 |

| 1.3 Develop and utilize potential Consistency ratio of judgment matrix: 0.0000 The weight of the overall goal: 0.2970 | | | |
|---|---|---|---|
| | Recoverable potential | Recoverable potential | $W_i$ |
| Recoverable potential | 1.0000 | 9.0000 | 0.9000 |
| Fluid salinity | 0.1111 | 1.0000 | 0.1000 |

The weights of evaluation indicators for deep geothermal resource zoning are shown in Table 7.

**Table 7.** The weights of the evaluation indicators.

| Evaluation Indicators | Weight |
|---|---|
| Buried depth of geothermal reservoir | 0.0223 |
| Thickness of geothermal reservoir | 0.1021 |
| Temperature gradient of cap rock | 0.0390 |
| Sedimentary characteristics | 0.2912 |
| Porosity of geothermal reservoir | 0.1602 |
| Sand–mud ratio | 0.0882 |
| Recoverable exploitation potential | 0.2673 |
| Salinity of geothermal fluid | 0.0297 |

(4) The calculation of a comprehensive evaluation value

In this study, the analytic hierarchy process (AHP) and multi-source information superposition evaluation method based on GIS were applied to the zoning evaluation of the hydrothermal geothermal target area. Through systemic analysis of hydrothermal geothermal influence factors, and according to the comprehensive weight of each influence factor, GIS was used to prepare a single-factor information map. Then, the investigation area was meshed; as shown in Figure 4, the grid map and the single-factor information map were assigned as overlays through the spatial analysis function of GIS to extract the value of each layer. Thus, the assignment in the graph corresponds to the corresponding grid points.

Each single-factor information map was registered and processed to form a composite superimposed evaluation model, and then the zoning evaluation map of the study area was produced. Formula (1) was used for GIS spatial analysis and evaluation:

$$P = \sum\nolimits_{n=i} P_i A_i \ (i = 1, 2, 3 \dots \dots n) \tag{1}$$

where $P$ is the comprehensive evaluation value for the zoning of hydrothermal geothermal resources in the evaluation unit, $n$ is the total number of the evaluation factors, $P_i$ is the score given by the $i$th evaluation index, and $A_i$ is the weight of the number $i$ evaluation index.

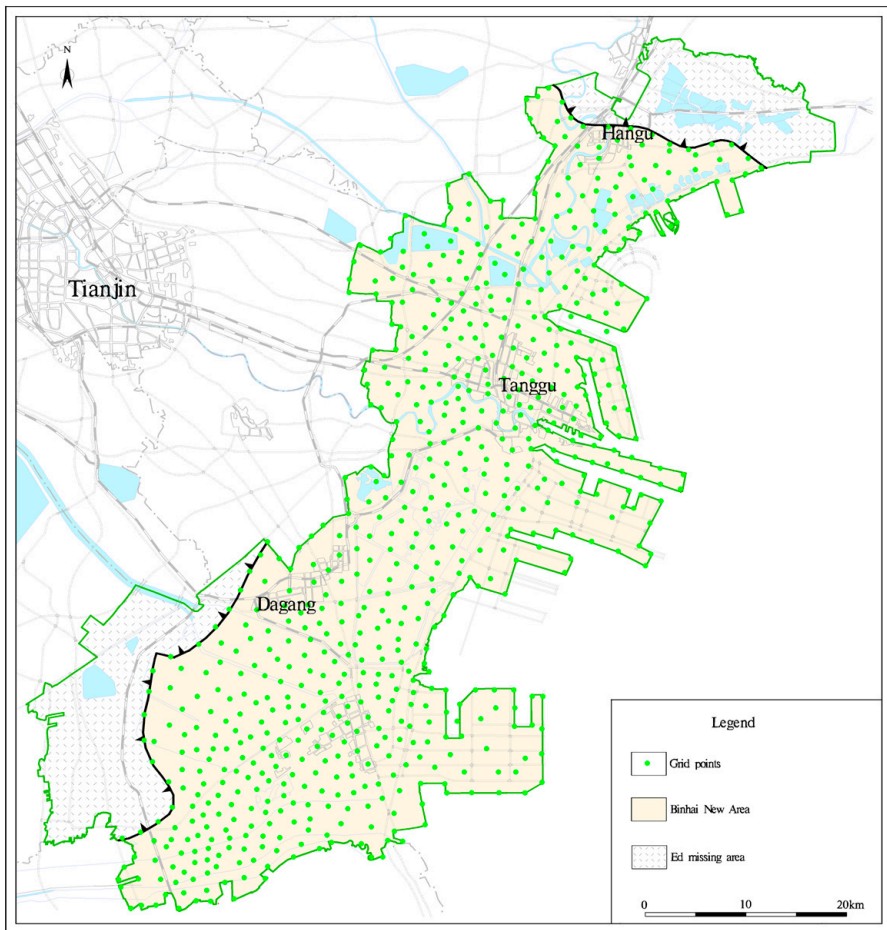

**Figure 4.** The evaluation grid points of the Dongying geothermal reservoir.

## 4. Results

The geothermal fluid is stored in deep underground geology formations with sufficient porosity and permeability, and its storage, migration, development and utilization are restricted by regional geology tectonics and geothermal reservoir conditions [17]. The evaluation area was meshed, and the map layers were extracted and assigned (Table 5) using the GIS spatial analysis function. Then, according to the weight of each evaluation index (Table 7), the comprehensive score value on each grid point was calculated by using the comprehensive index. By applying Formula (1), the evaluation map (Figure 5) of geothermal resource zoning was obtained. As can be seen from the Figure 5, in the class I area, the reservoir thickness of the Dongying formation is relatively large, and it belongs to the delta front sedimentation. It has good conditions of stratum storage, great thickness of geothermal reservoir, and enough resource exploitation potential. It is very suitable for the exploration and development of the Dongying geothermal resources. In the class II area, there are better conditions such as formation storage and sedimentary characteristics. It is more suitable for the exploration and development of geothermal resources. The class III area belongs to the delta plain deposit and has a general resource development prospect. In the class IV area, the thickness of heat storage is general, the condition of resource exploitation and supply is poor, and the prospect of resource exploitation is poor. The class V area has poor prospects for development due to poor formation storage and water-carrying capacity, as well as poor conditions for resource extraction and replenishment.

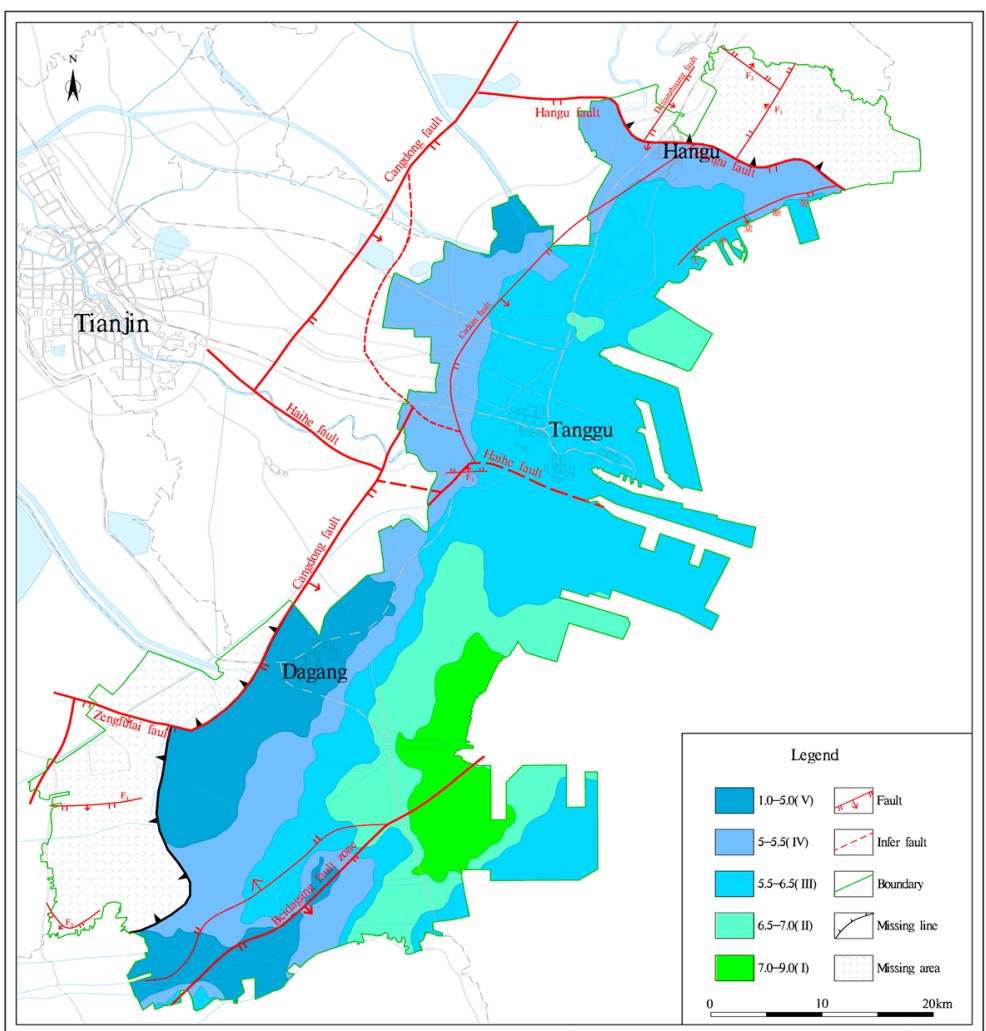

**Figure 5.** Comprehensive evaluation score of the Dongying geothermal reservoir.

## 5. Discussion

Finally, according to the evaluation results, the geothermal reservoir was divided into three levels of prospect target areas: the areas with an evaluation score over 6.5 were classified as the excellent prospect target areas; the areas with an evaluation score over 5.5 and less than 6.5 were classified as the general prospect target areas; and the areas with an evaluation score less than 5.5 were classified as the bad prospect target areas. The Beidagang wetlands and other nature reserves are included in the study area. According to the relevant regulations for nature reserves, geothermal well development and construction are prohibited. At the same time, there are oil-rich areas of the Dongying formation in the study area, and the geothermal fluid is rich in oil and gas, so it is not suitable for geothermal exploitation. Therefore, the oil-rich areas of the Dongying formation and the nature reserves in the study area (Figure 6) are regarded as the limiting conditions for geothermal exploitation. By superimposing the results of the first-level evaluation and the second-level evaluation, a results map of the Dongying geothermal resource exploration and development prospect target area was obtained (Figure 7). See Table 8 for the evaluation units included in the three levels of prospect target areas.

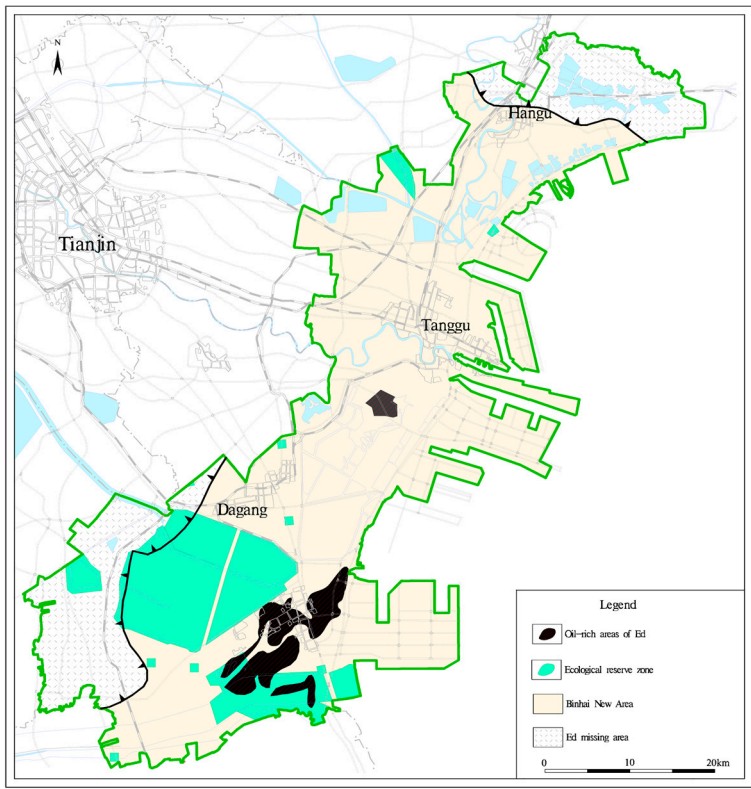

**Figure 6.** Distribution map of oil-rich areas and nature reserves.

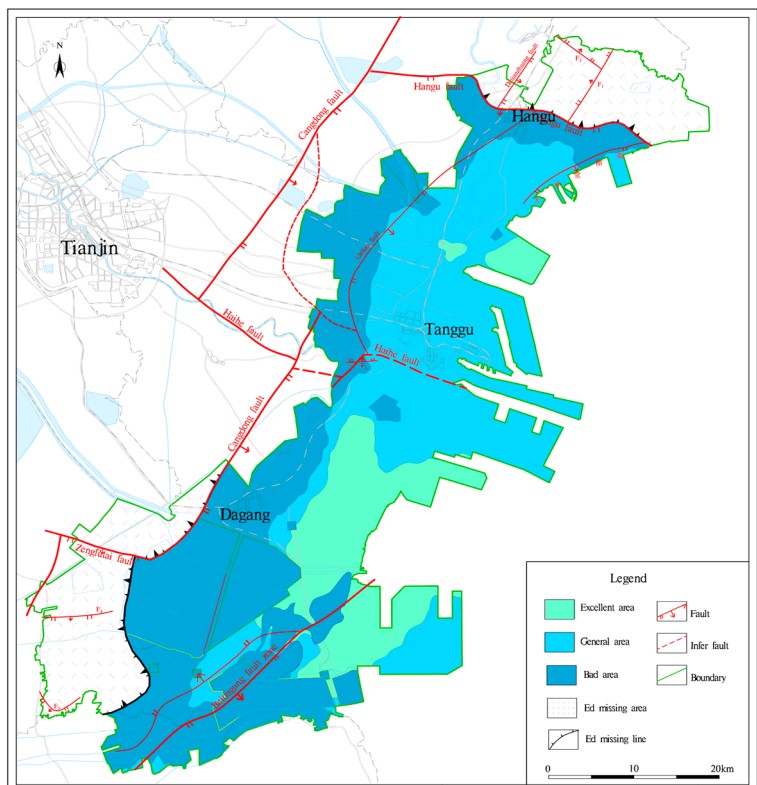

**Figure 7.** Prospective target areas of the Dongying geothermal reservoir.

**Table 8.** Optimization selection for the prospect target area of the Dongying geothermal reservoir in Tianjin Binhai New Area.

| Zoning | Target Area Location | Distribution Area (km$^2$) |
|---|---|---|
| Excellent prospect target area | Tianjin Lingang industrial zone–the northern area of Tianjin Nangang industrial zone | 314.33 |
| General prospect target area | Caijiabao–Dongjiang port–Tanggu urban–Lingang industrial zone–Lvjuhe village | 745.77 |
| Bad prospect target area | Dagang urban–Beidagang reservoir–Taiping town–Shajingzi village | 879.31 |

As can be seen from Figure 6, due to regional differences in the geothermal reservoir characteristics, the exploration and development prospects are not the same. According to the favorable distribution area statistics of the Dongying geothermal reservoirs at all levels (Table 8), the excellent prospect target area is 314.33 km$^2$, the general prospect target area is 745.77 km$^2$, and the bad prospect target area is 879.31 km$^2$. The excellent prospect target areas of the Dongying geothermal reservoir are mainly located in the southeast of the study area. Some are located in Tianjin Lingang industrial zone and the northern area of Tianjin Nangang industrial zone. The Dongying formation in this region is a delta front deposit with a large thickness, good conditions of geothermal fluid accumulation, and good resource potential. It is suitable for the development and utilization of geothermal resources in the Dongying formation. The general prospect target areas of the Dongying geothermal reservoir are mainly located in the northeast of the study area. They are located in the areas of Caijiabao, Dongjiang port, Tanggu urban area, Lingang industrial zone, and Lvjuhe. The bad prospect target areas of the Dongying geothermal reservoir are mainly located in the west and south of the study area. They are located in the areas of Dagang urban area, Beidagang reservoir, Taiping town, and Shajingzi. The Dongying formation in this region is a thin delta plain deposit, and the Cangdong fault is the water-resisting fault to the Dongying geothermal reservoir, so the resource exploitation and supply condition is poor. In addition, the oil-rich areas of the Dongying formation and the nature reserve zones in the south of the study area are also bad prospect target areas.

From the perspective of resource utilization and sustainable development, in order to maximize the availability of geothermal resources for economic construction services, the reasonable planning and development of geothermal resources can greatly supplement the consumption of energy resources. This provides grounds for the sustainable development and utilization of the Dongying geothermal resources in Tianjin Binhai New Area. Under the current situation, the large-scale and sustainable development and utilization of geothermal energy is part of the implementation of General Secretary Xi Jinping's National Energy Security Strategy, which is a response to global climate change. Implementing that strategy requires energy conservation and emission reduction via concrete measures to help achieve the goal of a "2030 carbon peak and 2060 carbon neutrality" [44].

## 6. Conclusions

According to the geothermal geology conditions, this paper established a geothermal resource zoning evaluation system and a multi-source information superposition evaluation method based on GIS. The following results were achieved:

(1) The development potential of the Dongying geothermal resources in Tianjin Binhai New Area was calculated using the Theis model of two-dimensional groundwater seeping. It was calculated that the recoverable capacity of geothermal fluid is $233.6 \times 10^4$ m$^3$/a under the single-well model and $315.36 \times 10^4$ m$^3$/a under the reinjection rate of 35% in the double-well model. The recoverable heat of this layer is $5.825 \times 10^{13}$ kJ, 18.47 MW in the single-well model, and it is $7.864 \times 10^{13}$ kJ, 24.93 MW in the double-well model. The geothermal resource reserves belong to the medium scale.

(2) The exploration and development target areas of the Dongying geothermal resources in Tianjin Binhai New Area were delineated on a regional scale through the quantitative zoning evaluation method. According to the statistics of the distribution area of favorable areas of the Dongying geothermal reservoir at all levels, the geothermal reservoir distribution area of the excellent prospect target area is 314.33 km$^2$, the general prospect target area is 745.77 km$^2$, and the bad prospect target area is 879.31 km$^2$.

(3) By using the analytic hierarchy process (AHP), the index of geothermal exploration and development regionalization was quantified. Combined with geothermal geology conditions and development constraints, the grade-two evaluation system was established for the delineation of geothermal resource exploration and development prospect areas. The quantitative zoning evaluation method can provide a reference for the optimization of the exploration and development target area of geothermal resources with low prospecting accuracy in key areas of China.

**Author Contributions:** J.L. provided project administration and was involved in the writing, methodology selection, the investigation and data collection; S.H. and H.X. helped in editing the manuscript; D.Y. helped with the methodology selection; F.Y. helped with conclusion and participated in editing the earlier versions of the manuscript. All authors have read and agreed to the published version of the manuscript.

**Funding:** This research was funded by the China Geological Survey, grant numbers DD20230077 and DD20230018.

**Institutional Review Board Statement:** Not applicable.

**Informed Consent Statement:** Not applicable.

**Data Availability Statement:** Data sharing is not applicable.

**Conflicts of Interest:** The authors declare no conflict of interest.

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
