# Peer review of "The Resource Potential and Zoning Evaluation for Deep Geothermal Resources of the Dongying Formation in Tianjin Binhai New Area"

_sustainability, doi:10.3390/su151612357_

Round 1

Reviewer 1 Report

This work focuses on comprehensive evaluation of resource potential and reliable basis of planning and management of geothermal utilization. Comments and suggestions are given below.

(1)  The abstract should be reorganized and shortened. The description for the background is lengthy.

(2)  The innovation of this work should be further addressed.

(3)  The language should be edited by a native English speaker.

(4)  The references should be widely and deeply reviewed and summarized. The literature review lacks critical comments. What is the research gap in the previous studies?

(5)  The method should be described in details.

(6)  The conclusions should be shortened. Please provide more specific findings or results.

Reviewer 2 Report

Remarks concerning the paper’s subject and idea

·       The feasibility of geothermal resource exploration depends not only on individual properties (Table 6) but also (and mostly) on their synergy, which is naturally quantified by flow and heat transfer models. Instead, the authors employed an approach with a one- or two-well model. The following essential factors are neglected: simulation heat transfer in watered rocks, (likely) lowering of the temperature during a long operation period due to cooling of rocks, and paying no attention to the deep geothermal flux that makes this resource renewable. These factors have to be considered in a really comprehensive approach. For this reason, the authors are recommended to refer to the other approaches on the international level; please see papers in Geotherm Energy like “Busby and Terrington, (2017) 5:7 DOI 10.1186/s40517-017-0066-z” and those of other researchers; the comparison should demonstrate the advantages of the method proposed in the submitted paper.

·       It is recommended to provide more details on how the weights in Table 6 have been calculated. It seems quite curious why sedimentary characteristics contribute 7.5 times more to geothermal resource assessment than the temperature of a geothermal reservoir. The analytical hierarchy process is, of course, a powerful tool, but it does not consider the physics and hydrodynamics of geothermal resource exploitation. Therefore, more details would be helpful about how the weights in Table 6 were calculated in terms of thermal fluid flow. More substantiated reasons/justifications of applicability to geothermal assessments are needed. No factor analysis or Design of Experiment has been provided in the text that could confirm the values of weights and factor significance.

·       It is unclear how the “recoverable exploitation coefficient” in Table 5 is graded as “weak”, “medium”, and “strong”. Numerical assessments, or the range of this parameter, are required to avoid subjectivity. It would be good to provide a physically correct definition of this parameter.

·       The main result of the paper is likely presented in Table 7. The distribution of areas by the zones of different exploitation conditions is single figures without ranges, which indicates that the uncertainty of parameters was likely neglected. It would be much sounder in terms of scientific relevance of the result to estimate how the parameter uncertainty may change the distribution in Table 7 or estimate result variability.

Remarks concerning some technical and editorial issues

·       Figure 2 demonstrates the vertical profile of rocks to a depth of about 450 m, whereas the calculations have been performed for depths of 2000 m (Table 1). Is Figure 2 representative for estimates related to deeper strata?

·       The calculation is based on three existing wells (line 162), but they are not demonstrated in maps; this is recommended for clarity.

·       Check for white spaces in the references, before citations in brackets [], and throughout the text. The inclusion of doi is recommended for the references. Please also check the correctness of references with “et al.”; this is likely to be included in citations but not in the reference list.

·       Improve the text readability in Figure 3.

·       Table 5: Details of the assessment method are required (grid step, range of parameters). Check the range “2.5–3.5 °C” of the temperature of the geothermal reservoir; the temperature should be likely higher.

·       Shortening the abstract is recommended, focusing on just what the authors have been done within this study.

Remove repetitions in the text (lines 15­16, 47–48, 211–213 etc.). Use language tools and ask for a review by a language expert to polish the text.

Reviewer 3 Report

The paper is about the potential of using deep geothermal energy in China. The paper presents a review of papers which bring information about the potential of geothermal energy use. The map for the region with detected potential is presented. Overal merit and scientific significance is low, but sufficient to be published. The results can be useful for further analises and further investigations.
I have some minor comments that could improve the quality of the paper:

1. Abstract is too long and to complex. Please rethink how to make it more concise. Please focus only on the main findings. Please highlight the strenghts of the paper and briefly introduce the method. 

2. Please improve the literature review. Please highlight the novelty of your paper at a basis of the previoius studies from other research gropus. The first point of the paper should be extended.

3. In discussion please describe better the connection between your paper and sustainability.

I have no comments.

Round 2

Reviewer 1 Report

This manuscript has been significantly improved.

Author Response

Deal reviewer,

     Our manuscript has been revised again, and submitted. Please you make valuable comments, thank you very much.

Reviewer 2 Report

The authors have partially responded to the remarks and slightly improved the quality, but some issues remain that have not addressed properly. Details see below.

1.       Partially corrected. The added explanation is based on insufficient data and the correctness of a simple model, and ignores other approaches to assessing geothermal potential.

2.       Actually, not corrected. It remains unclear how the coefficients in Table 6 were calculated. There are only qualitative arguments without quantitative assessment.

3.       Not corrected, no quantitative scale is added for the recoverable potential.

4.       Corrected.

5.       Corrected.

6.       Not corrected. Figure 3 looks the same as it did in the first version.

7.       Partially corrected (Figure 5), but no details about the assessment method have been added.

8.       Corrected.

9.       The sentences indicated were corrected, but still there are text fragments that need to be stylistically improved.
